# A Study of the Different Strains of the Genus *Azospirillum* spp. on Increasing Productivity and Stress Resilience in Plants

**DOI:** 10.3390/plants14020267

**Published:** 2025-01-18

**Authors:** Wenli Sun, Mohamad Hesam Shahrajabian, Na Wang

**Affiliations:** National Key Laboratory of Agricultural Microbiology, Biotechnology Research Institute, Chinese Academy of Agricultural Sciences, Beijing 100086, China; hesamshahrajabian@gmail.com (M.H.S.); wangna@163.com (N.W.)

**Keywords:** *Azospirillum*, *Bacillus*, biostimulant, plant growth-promoting rhizobacteria

## Abstract

One of the most important and essential components of sustainable agricultural production is biostimulants, which are emerging as a notable alternative of chemical-based products to mitigate soil contamination and environmental hazards. The most important modes of action of bacterial plant biostimulants on different plants are increasing disease resistance; activation of genes; production of chelating agents and organic acids; boosting quality through metabolome modulation; affecting the biosynthesis of phytochemicals; coordinating the activity of antioxidants and antioxidant enzymes; synthesis and accumulation of anthocyanins, vitamin C, and polyphenols; enhancing abiotic stress through cytokinin and abscisic acid (ABA) production; upregulation of stress-related genes; and the production of exopolysaccharides, secondary metabolites, and ACC deaminase. *Azospirillum* is a free-living bacterial genus which can promote the yield and growth of many species, with multiple modes of action which can vary on the basis of different climate and soil conditions. Different species of *Bacillus* spp. can increase the growth, yield, and biomass of plants by increasing the availability of nutrients; enhancing the solubilization and subsequent uptake of nutrients; synthesizing indole-3-acetic acid; fixing nitrogen; solubilizing phosphorus; promoting the production of phytohormones; enhancing the growth, production, and quality of fruits and crops via enhancing the production of carotenoids, flavonoids, phenols, and antioxidants; and increasing the synthesis of indoleacetic acid (IAA), gibberellins, siderophores, carotenoids, nitric oxide, and different cell surface components. The aim of this manuscript is to survey the effects of *Azospirillum* spp. and *Bacillus* spp. by presenting case studies and successful paradigms in several horticultural and agricultural plants.

## 1. Introduction

Plant growth-promoting rhizobacteria (PGPR) are a group of bacteria which reside in the root zone and are able to positively affect plant growth [1,2,3,4,5]. PGPR stimulate plant growth through a variety of functions, including soil structure formation; recycling of essential elements; organic decomposition; production of true growth regulators; dissolution of mineral nutrients; stimulation of plant growth; decomposition of organic pollutants; biological control of plant and soil pathogens; stimulation of root growth; and activation of abiotic stress resistance mechanisms such as salt resistance, drought resistance, resistance to heavy metals and different pollutants at high concentrations, and low temperature tolerance [6,7,8,9,10]. PGPR include the genii of *Aeromonas*, *Acinetobacter*, *Allorhizobium*, *Agrobacterium*, *Azoarcus*, *Arthrobacter*, *Azospirillum*, *Azorhizobium*, *Bradyrhizobium*, *Bacillus*, *Caulobacter*, *Burkholderia*, *Delftia*, *Chromobacterium*, *Frankia*, *Flavobacterium*, *Enterobacter*, *Mesorhizobium*, *Klebsiella*, *Paenibacillus*, *Micrococcus*, *Rhizobium*, *Pseudomonas*, *Thiobacillus*, *Streptomyces*, and *Serratia* [11,12,13,14,15,16,17,18,19].

*Azospirillum* is a genus of plant growth-promoting bacteria which has been extensively studied for decades [20,21,22,23,24]. Inoculation with *Azospirillum* is well known in fixing atmospheric nitrogen, and it can also provide plants with phytohormones (such as indole-3-acetic acid) and improve their tolerance to biotic and abiotic stresses [25,26,27]. *Azospirillum* is able to increase plant growth under abiotic stresses using different mechanisms, such as the production of phytohormones, osmotic adjustment, the stimulation of antioxidants, and defense strategies like pathogen-related gene expression [28,29,30]. Tarrand et al. (1979) reported that *A. lipoferum* CRT1 could stimulate pre-germinating or defense events, increase surface bacterial counts, lower contents of energetic primary metabolites, and improve the root surface area and photosynthetic yield in three-leaf plantlets [31]. The damaging impacts of NaCl on wheat (*Triticum aestivum* L.) seedlings were decreased by inoculation with *Azospirillum* strains, and the inoculation could also induce an increase in the dry weight and grain yield of shoots under severe water salinity [32,33,34,35,36,37].

Tarrand, Krieg and Döbereiner (1978) reported that *A. brasilense* possessed special genes for bacteriophytochrome which could control carotenoid-independent reactions to photodynamic stress [38]. This proves the fact that, even though the bacteria are not phototrophic, they are equipped to sense and react to light [39]. *Azospirillum* spp. have been studied for a long time as model organisms to study mutualistic interactions between bacteria and plants, as they can improve plant growth by producing phytohormones such as indole-3-acetic acid (IAA). The diversity of the aldehyde dehydrogenases (ALDHs) that they possess can significantly influence their ability to produce IAA [40,41]. Different strains of *Azospirillum* have a special ability to compete for colonization sites in the upper and lower soil regions of crops, and increase synthesis of phytohormones [42]. This review article aims to survey the effects of *Azospirillum* by presenting case studies and successful paradigms in different horticultural and agricultural crops.

This research examined the scientific literature from 1990 to December 2024 by conducting a bibliometric analysis of the literature published on the Web of Science database, including more than one thousand articles. The information provided was obtained from randomized control experiments, review articles, and analytical observations and studies which were gathered from various literature sources such as PubMed, Science Direct, Scopus, and Google Scholar. The keywords used were the Latin and common names of different agricultural and horticultural species, microbial biostimulants such as *Azotobacter*, *Fusarium*, biostimulants, phytohormones, and plant growth-promoting rhizobacteria. The most important benefits of *Azospirillum* spp. are presented in Figure 1.

## 2. Plant Growth Promotion

Plant growth-promoting rhizobacteria (PGPR) positively influence the development and growth of plants [43], which is an important characteristic of these bacteria [44]. Their direct growth stimulation mechanisms are related to improving the absorption of nutrients and regulating and synthesizing plant hormones [45,46]. Their indirect influence consists of a wide range of mechanisms which may suppress or prevent plant diseases [47]. Different microorganisms such as *Burkholderia* [48], *Azotobacter* [49], *Rhizobium* [49], *Pantoea* [50], *Enterobacter* [51], *Pseudomonas* [52], *Bacillus* [52], *Microbacterium*, *Micrococcus* [53], *Stenotrophomonas* [53], and *Serratia* [54] have been found to be wonderful agricultural growth-stimulating agents. Some microorganisms have been applied as microbial inoculants and bioinoculants to enhance crop productivity without causing contamination [55,56]. They secrete phytohormones such as gibberellins, cytokinins, and auxins which can induce changes in plant root architecture, and promote the development of adventitious roots [57,58].

*Azospirillum* is of the main characterized genii of plant growth-promoting rhizobacteria and belongs to the class of *Alphaproteobacteria*, order *Rhodospirillales* [59,60]. Some of the most important isolated species of *Azospirillum* are *A. largimobile* (Skerman et al., 1983) in grass [61], *A. oryzae* (Ahlb.) (Cohn, 1884) in rice (*Oryza sativa* L.) [62], *A. lipoferum* (Tarrand et al., 1979) in wheat [63], *A. irakense* (Khammas et al., 1991) in rice [64], *A. formosense* (Lin et al., 2012) in rice [65], *A. thiophilum* (Lavrinenko et al., 2010) in water [66], *A. griseum* (Yang et al., 2019) in Agua (*Trichantera gigantean* Nees) [67], *A. oleicasticum* (Wu et al., 2021) in oil [68], *A. rugosum* (Young et al., 2008) in contaminated soil [69], *A. picis* (Lin et al., 2009) in tar [70], *A. fermentarium* (Lin et al., 2013) in fermenter [71], *A. humicireducens* (Zhou et al., 2013) in microbial fuel cell [72], *A. brasilense* (Tarrand et al., 1979) in grass [73], *A. halopraeferens* (Reinhold et al., 1987) in grass [74], *A. doebereinerae* (Eckert et al., 2001) in grass [75], *A. melinis* (Peng et al., 2006) in grass [76], *A. canadense* (Mehnaz et al., 2007) in corn (*Zea mays* L.) [77], *A. zeae* (Mehnaz et al., 2007) in corn [78] *A. palatum* (Zhou et al., 2009) in soil [79], *A. soli* (Lin et al., 2015) in agricultural soil [80], and *A. agricola* (Lin et al., 2016) in agricultural soil [81]. It has been reported that *Azospirillum* can assist plant growth under challenging conditions such as drought, salinity, and nutrient-limited conditions through the production of different osmolytes and improved water intake [82,83,84]. Peng et al. [85] reported that *A. brasilense* improved the chlorophyll content and growth of *Chlorella sorokiniana*, and mitigated oxidative stress. Inoculation of different plants with *Azospirillum* strains could have positive effects under various stress conditions. For example, *A. brasilense* increased the fresh weight of shoots under cadmium stress in thale cress (*Arabidopsis thaliana* L.) [86], and increased the diameter and seed yield of rosettes under drought stress [87]. *A. lipoferum* enhanced the root elongation and root biomass of barley (*Hordeum vulgare* L.) under cadmium stress [88], and *A. brasilense* increased total plant weight under salinization [89]. *A. brasilense* increased the root weight and root length of cucumber (*Cucumis sativus* L.) under copper stress [90], and increased the dry weight and the number of leaves of flax (*Linum usitatissimum* L.) plants under salinization [91]. Both *A. brasilense* and *A. lipoferum* were shown to increase the total biomass and plant height of corn plants under drought stress [92,93]. *A. brasilense* increased the total biomass as well as root and shoot biomass of pak choi (*Brassica rapa* subsp. *Chinensis*) plants under cadmium stress [94,95]. Inoculation with *A. brasilense* also improved the final yield of tomato plants [96] and wheat plants [97,98]. It has also been reported that the application of *A. brasilense* could enhance shoot height and root length of white clover (*Trifolium repens* L.) plants [99]. *Azospirillum* has also played a role in plant defense against stress factors such as hydrocarbon pollution [100,101,102,103,104,105,106], heavy metal pollution [107,108,109,110,111,112], phytopathogenic infection [113,114,115,116,117], pesticide pollution [118], and osmotic stress [119,120]. The participation of *Azospirillum* in the plant, defense against stress factors is shown in Table 1.

## 3. *Azospirillum* spp. Benefits and Importance

*Azospirillum* is a member of the family *Rhodospirillaceae* in the order *Rhodospirillales,* belonging to the class *Alphaproteobacteria* [121,122,123,124,125,126,127,128]. *Azospirillum* species are mainly soil bacteria that coevolved with vascular plants [129]. They show a versatile N- and C-metabolism, which makes them well suited to establish in the competitive environment of the rhizosphere [130]. Different plant-associated *Azospirillum* species have been found to show direct nitrogen fixation, drought and salt stress alleviation, phosphate solubilization, reduction of agricultural environmental effects via nitrous oxide reduction, and promotion of root development [131,132,133,134,135,136,137]. Garcia et al. [138] reported that *Azospirillum brasilense* Az19 was a plant-beneficial bacterium capable of protecting plants from the adverse impacts of drought. It could fix nitrogen, but the major mode of action was phytohormone production, and its inoculation gave a mean grain yield response of 10% [139,140,141,142,143,144,145]. The genes related to its activities were found to be *pqq*, ACC deaminase (acds), *nif*, and “indole” acetic acid biosynthesis genes such as *ipdC*, *iaaM*, and *iaaH* [146]. Sharifsadat et al. [147] found that the activity of nitrate reductase and total nitrogen content was boosted after inoculation by *Azospirillum* in rice plants. Sharifsadat et al. [147] also reported that after inoculation with *Azospirillum* spp. modified using lipid peroxidation, the amount of hydrogen peroxide, NADPH oxidase, and the activities of ferulic acid peroxidase, total nitrogen, nitrate reductase, pectinase, xylanase, mannanase, and pectin methyl esterase were improved significantly. Koul and Kochar [148] observed that *A. baldaniorum* Sp245 was involved in the regulation of important bacterial biological networks, and that they can regulate biofilm formation, indole acetic acid production, and polyhydroxybutyrate synthesis. *A. brasilense* AbV5 and AbV6 were found to increase the tolerance of maize plants to abiotic and biotic stresses, and improve shoot fresh weight, root length and the content of chlorophyll b [149]. Seed inoculation with *A. brasilense* can promote nitrogen leaf concentration, root mass, nitrogen in grains, and grain yield [150]. *A. brasilense* has been considered to be an appropriate technology for stimulating plant–soil nitrogen management which can lead to more sustainable maize production [151]. Pereyra et al. [152] reported that *Azospirillum* increases water status in wheat seedlings under osmotic stress, and wider xylem vessels in the coleoptiles of *Azospirillum*-inoculated osmotic-stressed wheat seedlings has been reported. Foliar application with *Pseudomonas* sp. and *Azospirillum* sp. in glyphosate-treated plants promoted shoot and root biomass and increased phytohormone content, photosynthetic pigments, and maize yield [153]. A correlation between mutation-induced alterations in the lipopolysaccharides of *Azospirillum* and bacterial activity towards wheat roots has also been reported; lipopolysaccharides, which are revealed in the outer membrane of gram-negative bacteria, are also involved in interactions with plants [154]. In South America, *A. brasilense* Az39 isolated from roots of wheat, and its inoculation in maize plants was shown to result in a higher tolerance to osmotic and salt stress [155]. In one experiment, it was reported that the application of *A. brasilense* Sp245 had a positive influence on the node number and stem length of plum (*Prunus domestica*) and apple (*Malus* x *domestica*) fruits [156]. Gonzalez et al. [157] reported that *A. brasilense* can increase the root index, promote rhizogenesis, and reduce the undesirable impacts of NaCl in jojoba (*Simmondsia chinensis* (Link) C. K. Schneid.) rooting. *Azospirillum*-inoculated lettuce seeds have shown higher vegetative growth and germination compared to non-inoculated controls after being exposed to NaCl [158]. *A. brasilense* FP2 is considered to be an important plant growth-promoting bacterium in barley (*Hordeum vulgare* L.) [159]. The major characteristic of *Azospirillum* spp. is their capacity to release phytohormones, increase root growth, fix atmospheric nitrogen, and improve resistance to drought stress, mineral, and water uptake [160]. The production of phytohormones such as gibberellins, abscisic acid, and IAA (both in association with the plant and in the culture) is usually used to illustrate the impacts of *Azospirillum* spp. [161,162,163,164,165,166,167,168,169,170,171,172,173,174]. The effects of different species of *Azospirillum* are shown in Table 2.

## 4. Conclusions and Future Prospects

Biostimulants, a growing field in agriculture, hold the special potential to enhance plant growth, improve crop yields, boost resilience, and decrease the environmental effect of farming practices. Bacterial plant biostimulants are a major type of plant biostimulants which can colonize the plant rhizosphere, improve mineral and nutrient uptake in plants, control plant pathogens, enhance plant growth, and improve tolerance and resistance of plants to different types of abiotic and biotic stresses. Bacteria can interact with plants in a variety of ways due to their wide range of functions, such as their active roles in the supply of nutrients, biogeochemical cycles, improving nutrient consumer effectiveness, improving stress tolerance, induction of resistance, morphogenetic control, involvement in plant growth regulators, transient or lifelong associations, and the continuum of symbiosis. Bacterial plant biostimulants can increase productivity and improve plant growth through numerous mechanisms which include antimicrobial metabolites and different lytic enzymes; nutrient acquisition by nitrogen fixation; solubilization of insoluble minerals such as Zn, K, and P; siderophores; of course, organic acids; the action of growth regulators and stress-induced phytohormones; mitigating the adverse impacts of abiotic stress such as high soil salinity, drought, oxidative stress, extreme temperatures, and heavy metals using various modes of action; and plant defense induction procedures. The genus *Azospirilum* belongs to the *Rhodospirillaceae* family, which is basically constituted of aquatic genera. *Azospirillum* is capable of boosting plant growth under abiotic stresses using numerous mechanisms, such as osmotic adjustment, antioxidants, phytohormone production, and defense strategies like pathogen-related gene expression. The mode of action of *Azospirillum* is different depending on climate and soil conditions. The solubilization of minerals such as phosphorus and iron, and its growth promotion consists of trehalose, polyamine, and phytohormone production, as well as nitrogen fixation. In this review, we have clarified the importance of *Azospirillum* spp. in enhancing plant growth and increasing plant protection against negative environmental parameters. The incorporation of bacterial biostimulants in cropping systems has been revealed to be a promising technique for sustainable agriculture and ensuring food security. However, more research is needed on their mechanisms, especially the molecular procedures involved, considering parameters related to sustainable agricultural systems.

## Figures and Tables

**Figure 1 plants-14-00267-f001:**
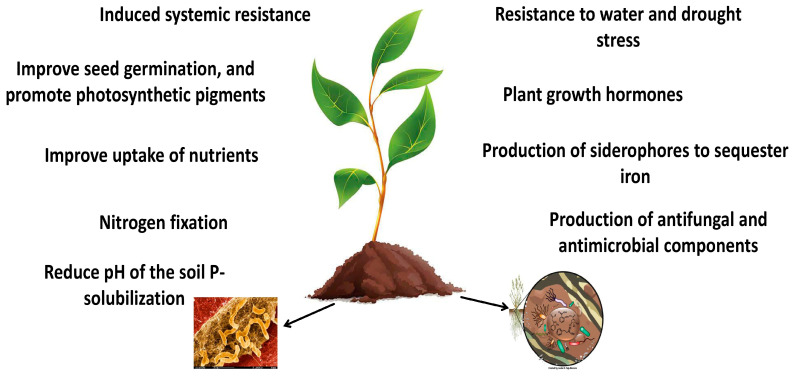
The most important effects of *Azospirillum* spp.

**Table 1 plants-14-00267-t001:** The participation of *Azospirillum* in plants, defense against stress factors.

Mechanism	*Azospirillum* Strains	Key Points	Reference
Hydrocarbon pollution	*A. brasilense*; *Azospirillum* sp.	Effective in microbial communities which can break down hydrocarbons.	[100]
	*A. brasilense* SR80; *A. brasilense* MT814302; *A. brasilense* MT814301; *A. brasilense* MT814300; *A. brasilense* AF411852; *Azospirillum* sp.	Biodegrade phenol, benzoate, and crude oil.	[101,102,103,104]
	*A. brasilense* strain 11; *A. brasilense* Az39; *Azospirillum* sp.	Some bacteria were found in biofilms which decompose hydrocarbons.	[105,106]
Heavy metal pollution	*Azospirillum* sp.; *A. brasilense*; *A. baldaniorum*	Tolerate lead, copper, cadmium, and arsenic.	[107,108,109]
	*Azospirillum* sp.; *A. brasilense*; *A. baldaniorum*	Significant effects on the content of photosynthetic pigments in corn in the presence of arsenic.	[109]
	*Azospirillum* sp.; *A. brasilense* Sp245; *A. baldaniorum*	Effective in reducing cadmium toxicity for barley, pak choi, and Arabidopsis.	[110]
	*Azospirillum* sp.; *A. brasilense*; *A. baldaniorum*	Decrease copper stress in wheat.	[111]
	*Azospirillum* sp.; *A. brasilense*; *A. baldaniorum* Sp245	Reduce copper stress in cucumber.	[112]
Infection of plants with phytopathogens	*A. brasilense* Sp245; *A. brasilense* Sp7; *Azospirillum* sp. BNM64	Able to biologically control phytopathogens.	[113,114]
	*A. brasilense*; *Azospirillum* sp. ERC2; *Azospirillum* sp. REC3	Induction of changes in the host plant metabolism, and the synthesis of siderophores which can limit the availability of Fe to phytopathogens.	[115]
	*A. brasilense*; *Azospirillum* sp.	Limit the development of phytopathogens via the induction of systemic resistance in plants.	[116,117]
Pesticide pollution	*Azospirillum* sp.	Degrade the pesticide Ethion.	[118]
Osmotic stress	*A. brasilense*; *Azospirillum* sp.	The main osmolytes are glycine-betaine, soluble sugars, and prolines which can be reduced through osmotic stress.	[119]
	*A. brasilense* Sp7; *Azospirillum* sp.	Can use osmoadaptation to increase growth and nitrogen fixation under salt stress conditions.	[120]

**Table 2 plants-14-00267-t002:** The effects of different species of *Azospirillum* on yield and yield components of various plants.

Plant	Plant Family	*Azospirillum* Species	Key Points	Reference
Arabidopsis (*Arabidopsis thaliana*)	Brassicaceae	*A. brasilense*	Changes root system architecture, leading to major transcriptional changes in nitrogen metabolism and carbon process. Improves stiffened cell walls and peroxidase activity.	[175]
		*A. brasilense*	Increases shoot fresh weight, seed yield, and rosette diameters under cadmium and drought stress.	[86,87]
		*A. brasilense* Sp245	Influences the growth of plants through a mechanism involving target rapamycin.	[176]
		*A. brasilense* Sp245	Increases yield and yield components.	[176]
Barley (*Hordeum vulgare*)	Poaceae	*A. lipoferum*	Increases root elongation and root biomass under cadmium stress.	[88]
Basil (*Ocimum basilicum* L.)	Lamiaceae	*A. brasilense* Sp245	Its benefits on the basis of *Azospirillum brasilense* Sp245 were significantly associated with the synthesis of phytohormones.	[112]
Candyleaf (*Stevia rebaudiana*)	Asteraceae	*A. brasilense*	Significant upregulation of genes accountable for the biosynthesis of steviol glycosides (*UGT76G1*, *UGT74G1*, *UGT85C2*, *Kaurene* oxidase, *entKO*)	[177]
Cherry pulm (*Prunus cerasifera* L.)	Rosaceae	*A. brasilense* Sp245	Promotes the rooting of explants.	[178]
Chickpea (*Cicer arietinum* L.)	Fabaceae	*A. lipoferum* FK1	Decreases the inhibitory effects of salinity through stress-related genes, antioxidant machinery, and modulating osmolytes.	[179]
		*A. brasilense* EMCC1454	Increases plant growth, and reduce chromium toxicity effects by modulating photosynthesis, antioxidant machinery, stress-related gene expression, and osmolyte production.	[180,181,182]
Common bean (*Phaseolus vulgaris* L.)	Fabaceae	*A. brasilense* CD	Decreases the negative effects of salt stress.	[183]
		*A. brasilense*	Positively influences shoot and root dry weight.	[184]
		*A. brasilense*	Increases grain yield, number of pods per pot, pod weight, and number of grains per pod.	[185]
Coriander (*Coriandrum sativum*)	Apiaceae	*A. brasilense*	Increases dry weight, fresh weight, total plant fresh weight, total plant dry weight under salinization.	[89]
Corn (*Zea mays* L.)	Poaceae	*A. brasilense* AbV5/AbV6	Seed inoculation can increase crude protein content, lead nitrogen content, starch content, and total sugar content of baby corn crops.	[186,187]
		*A. brasilense* Az39	Increases plant growth as it has a high amount of cytokinins, auxins, and gibberellins.	[188]
		*A. brasilense* AZ	Under water stress, it can promote maize root attributes.	[189,190,191]
		*A. brasilense* Az1 and Az2	Inoculation with it is an ecologically and economically viable technology.	[191]
		*A. argentinense* Az19	Prevents the negative impacts of water deficits, especially at the flowering stage, on maize growth.	[192]
		*Azospirillum* sp. Sp7	Increases the tolerance of seedlings to drought.	[193,194]
		*A. lipoferum* CRT1	Increases yield and yield components.	[195]
		*A. lipoferum* HM053	Stimulates photosynthesis and increases chlorophyll concentration.	[196]
		*A. brasilense*	Influences seedlings at the early stages, and ultimately influence the final growth.	[197]
		*A. brasilense* Ab-V5	Increases nitrogen use efficiency and improve biochemical characteristics.	[198]
Cucumber (*Cucumis sativus*)	Cucurbitaceae	*A. brasilense*	Under copper stress, it enhances root weight, root length, and root tips.	[90]
Sweet corn (*Zea mays* L. *Saccarata*)	Poaceae	*A. brasilense* (LB1-1, LB1-2, LB1-3, and LB1-4)	Significantly increases plant growth.	[199]
Cotton (*Gossypium hirsutum*)	Malvaceae	*A. brasilense*	Increases plant height, yield, total nitrogen content, and high biomass on cotton varieties (H-117, HD-123).	[200]
Cowpea (*Vigna unguiculata* (L.) Walp.)	Fabaceae	*A. brasilense* Ab-V5 and Ab-V6	Improves the growth of cowpea.	[201]
		*A. brasilense*	Increases plant biomass, grain yield, and photosynthetic pigments.	[202]
Cucumber (*Cucumis sativus* L.)	Cucurbitaceae	*A. brasilense* Cd (DSM-1843)	Decreases the stress signs caused by both the double Fe and Cu, Cu toxicity, and Cu deficiency, and improve the root system.	[90]
		*A. brasilense*	Modulates Fe acquisition in plants by differently triggering gene transcription.	[203]
		*A. brasilense* Sp245	It has been considered as a general plant root colonizer.	[204]
		*A. brasilense* Sp7, Sp7-S, and Sp245	Inoculated seedlings produce greater root biomass, longer roots, and higher total phosphorus content.	[205,206]
Flax (*Linum usitatissimum*)	Linaceae	*A. brasilense*	Increases root length, shoot length, dry weight of shoot, fresh weight of shoot, dry weight of root, and the number of leaves under salinization.	[91]
Hopbush shrub (*Dodonaea viscosa* L.)	Sapindaceae	*A. lipoferum*	Favorably affects plant growth parameters, stem length, root length, and stem fresh and dry weights.	[207]
Jojoba (*Simmondsia chinensis* L.)	Simmondsiaceae	*Azospirillum* sp. Az39	Induces jojoba rooting, rooting percentage, survival rate, and acclimatization.	[208]
Lettuce (*Lactuca sativa* L.)	Asteraceae	*A. argentinense* Az39	Dual inoculation of *Pseudmonas* strain and *Azospirillum* significantly influences plant growth, extended root survival, and increased chemical components.	[209]
		*A. lipoferum* CRT1	It has positive effects on seedlings growth.	[210]
		*Azospirillum* sp.	Seed inoculation with *Azospirillum* improves biomass and lettuce quality of plants under salt-stress conditions.	[211]
		*A. brasilense* AbV5 and AbV6	Increases cell membrane integrity index, net photosynthesis rate, relative water content, stomatal conductance, and total chlorophyll.	[212]
		*A. brasilense* AbV5 and AbV6	Increases the accumulation of K, P, N, Ca, Mg, B, S, Mn, Fe, and Zn in plants.	[213]
Lima bean (*Phaseolus lunatus* L.)	Fabaceae	*A. baldaniorum* Sp245	Under salt stress, its inoculation can attenuate the negative impacts of salt stress, improving the growth and symbiotic performance of lima bean.	[214]
Lisianthus (*Eustoma grandiflorum* (Raf.) Schinn.	Gentianaceae	*A. brasilense* Az39	Significantly increases leaf area, number of leaves, dry and fresh weight of seedlings, number and length of roots, leaves thickness, diameter of the vascular bundle, and root thickness.	[215]
Maize (*Zea mays*)	Poaceae	*A. brasilense; A. lipoferum*	Increases total biomass and plant height.	[92,93]
Olive (*Olea europaea* L.)		*A. baldaniorum* Sp245	Induces cellular activities and improve the rooting rate of cuttings.	[216]
Onion (*Allium cepa* L.)	Amaryllidaceae	*A. brasilense* 1224^T^	Significantly increases onion yield.	[217]
Pak choi (*Brassica chinensis* L.)	Brassicaceae	*A. brasilense*	Promotes antioxidant enzyme content, shoot biomass, and reduce Cd translocation factors.	[94,95]
Palisade grass (*Urochloa brizantha*)	Poaceae	*A. brasilense* CNPSo 2083 (Ab-V5) and CNPSo 2084 (Ab-V6)	Increases growth and development of seedlings.	[218]
Palmarosa (*Cymbopogon martinii*)	Poaceae	*A. brasilense*	Stimulates VAM colonization and promotes VAM spore population.	[219]
Pea (*Pisum sativum* L.)	Fabaceae	*Azospirillum* spp. Er-20	Increases photosynthetic pigments, chlorophyll a and b, total carotenoids, total phenolics, and chlorophyll concentrations.	[220]
Pepper (*Capsicum annuum* L.)	Solanaceae	*Azospirillum* spp.	Supply a notable amount of nitrogen to pepper seedlings.	[221]
		*A. brasilense*	Increases potential availability of nutrients for uptake, particularly for fruit quality characteristics.	[222]
Proso millet (*Panicum miliaceum*)	Poaceae	*A. brasilense* RAU-1 and RAU-2	Significantly boosts the uptake of Fe and total yield.	[223]
Potato (*Solanum tuberosum* L.)	Solanaceae	*A. lipoferum* AL-3	Helps potato plants resist against blight disease via induced systemic resistance as well as induced to increase the quantity of total phenolics, and defense-related enzymes such as polyphenol oxidase, peroxidase, and phenylalanine ammonia lyase.	[224]
		*Azospirillum* spp.	Improves nitrogen use efficiency and enhance plant growth.	[225]
Purple basil (*Ocimum basilicum* L.)	Lamiaceae	*A. baldaniorum* Sp245	Strong correlation with the synthesis of phytohormones.	[112]
Radish (*Raphanus sativus* L.)	Brassicaceae	*A. brasilense* Cd DMS 1843	Responsible for improving and activating some physiological mechanisms of the plant.	[226]
Rice (*Oryza sativa* L.)	Poaceae	*Azospirillum* spp. Az2 and As 5	*Azospirillum* spp. indicates notable higher nitrogen fixation and N_2_-ase activity.	[227]
		*Azospirillum* spp.	Combined application of *Azospirillum* spp. and *Azotobacter* improves development and the growth of rice.	[227]
		*Azospirillum* sp. B510	Increases nitrogen uptake and plant growth, and it can be considered as a key solution for chemical-free sustainable agriculture.	[228]
		*A. lipoferum* 4B	Induces the improvement of plant secondary metabolites.	[229]
		*Azospirillum* sp. B510	Induces disease resistance in rice, caused by *Magnaporthe oryzae*.	[230,231]
		*Azospirillum* sp. B510	Influences the bacterial community structure and increases transcriptomic response in roots and shoots.	[232]
		*A. brasilense* Ab-V5 and Ab-V6	Combined with nitrogen fertilizer, it enhances the dry mass of the aerial part of rice and grain yield.	[233]
		*A. brasilense; A. irakens*	Increases total nitrogen and activity of nitrate reductase content.	[147]
		*Azospirillum* sp. B510	Significantly controls and improves root growth.	[234]
Ryegrass (*Lolium perenne* L.)	Poaceae	*A. brasilense* D7	Increases plant growth through volatile organic compounds.	[235]
Sorghum (*Sorghum bicolor* L.)	Poaceae	*A. brasilense* SM	Beneficially and positively influences the growth of sorghum.	[236]
		*A. brasilense*	Has a positive effect on root development, and the probable role of auxin in this process.	[237]
		*A. brasilense*	It can be applied as a nitrogen fertilization strategy, and improved dry matter production.	[238]
Soybean (*Glycine max* (L.) Merr.)	Fabaceae	*A. brasilense*	Its inoculation or co-inoculation can increase seed protein and plant growth of plants.	[239]
		*A. brasilense* Ab-V5 and Ab-V6	Promotes grain yield and growth parameters and mitigatesthe impacts of water stress on plants.	[240]
		*Azospirillum* spp.	Increases root biomass, and improved proline content.	[241]
		*A. brasilense*	Increases nodulation, grain yield, and nitrogen fixation.	[242,243]
		*A. brasilense* Az39	Safely and appropriately increases growth and yield of soybean exposed to As.	[244]
		*A. brasilense* Ab-V5 and Ab-V6	Increases grain yield and nodulation.	[245]
Strawberry (*Fragaria ananassa*, Duch.)	Rosaceae	*A. brasilense* REC3 and PEC5	Leads to better growth which can contribute to a sustainable agricultural practice.	[246]
Sweet-potato (*Ipomoea batatas* (L.) Lam.)	Convolvulaceae	*A. brasilense*	Positive influence on root yield, provides beneficial impacts on plant root association.	[247]
Sugarcane (*Saccharum* spp.)	Poaceae	*A. brasilense*	Increases plant cane and ratoon as well as stalk yield and stalk production.	[248]
		*A. brasilense*	Increases sugarcane productivity at the tillering and sprouting stages with high potential to improve economic and agronomic benefits.	[248]
Sweet leaf (*Stevia rebaudiana* (Bertoni)	Asteraceae	*A. brasilense*	Increases the physio-biochemical and growth of plants.	[178]
Tomato (*Solanum lycopersicum* L.)	Solanaceae	*A. brasilense* BNM65	Induces higher leaf area and total biomass.	[249]
		*Azospirillum* sp. B510	Activates the innate immune system against bacterial leaf spot.	[250]
		*A. brasilense* (DSM 1843, Leibniz-Institute DMSZ, Braunschweig, Germany)	Combined application with solarized manure improves root length, growth emergence, final yield, protein, and lipids in plants.	[251]
		*A. brasilense*	Increases root biomass under salinization.	[96]
Wheat (*Triticum aestivum* L.)	Poaceae	*A. brasilense* Sp245	Positive effect on yield and yield components.	[252,253]
		*A. brasilense* Sp245	Increases coleoptile length, root surface, and dry and fresh weight.	[254]
		*A. brasilense* Sp245	Protects seedlings from water deficiency through changes in fatty acid in roots.	[255]
		*A. brasilense* Az39	Reduces plant damages caused by stresses.	[256]
		*A. brasilense* Az39	Reduces Cd entrance into wheat roots and reduces Cd/Fe imbalance.	[256]
		*Azospirillum* spp.	Increases tolerance to salinity, and influences proline accumulation, photosynthetic pigment contents, and uptake of water.	[257,258,259]
		*Azospirillum* spp. INTA Az-39	Shows more vigorous vegetative growth, with higher root and shoot dry matter accumulation.	[260,261]
		*A. brasilense*	Makes partial biological nitrogen production possible.	[262]
		*A. brasilense*	Results in higher grain yield, higher number of grains per spike, and higher crop growth rate.	[263]
		*A. brasilense* EPS	Induces root growth.	[264,265,266,267]
		*A. brasilense*	Improves a thousand grain weight, grain yield per plant, plant height, spike length, and the number of grains and spikelets per spike under Arsenic stress.	[97]
		*A. lipoferum*	Increases final yield under drought stress.	[98]
White clover (*Trifolium repens*)	Fabaceae	*A. brasilense*	Increases root length and shoot height under salinization.	[99]
Durum wheat (*Triticum durum*)	Poaceae	*A. brasilense*	Stimulates the growth of the plant, the length of roots and leaves and chlorophyll.	[268,269,270,271,272]
Yellow lapacho (*Handroanthus ochraceus*)	Bignoniaceae	*A. brasilense* Cd and Az39	Leads to the highest root index, smaller number and size of stomata, and high development of dendritic trichomes.	[273,274]

## Data Availability

Not applicable.

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
