# Peer review of "A Study of the Different Strains of the Genus Azospirillum spp. on Increasing Productivity and Stress Resilience in Plants"

_plants, 2025, doi:10.3390/plants14020267_

Round 1
Reviewer 1 Report
Comments and Suggestions for Authors
The review article by Sun et al. focuses on the importance of Azospirillum application in agriculture.
The article needs majour revision, particularly:
-describe in detail how and when the databases were interrogated. The method is too briefly described at lines 71-72
-figure 1 should have a much longer legend
-table 1: avoid "It could" everywhere, just mention the effect, same for table 3
-table 2: also here the summary of the data should be more direct: for example "biodegradation of phenol, benzoate and crude oil" NOT "azospirillum strains can biodegrade phenol,...."This is more a sentence that would go in main text and not in a table
-the articles mentioned in paragraph 3 talk about benefits and importance of Azospirillum spp., isn't this the same topic developped in table 3. Why articles mentioned in the paragraph are not included in table 3? The differences or the link between text and table is not clear
-authors should introduce also a paragraph (or columns within the tables) mentioning how the bacteria were produced, delivered, and at what concentration. This is a key information missing and extremely usefull for potential readers
-lines 94 and subsequent the "in" is it because they have been studied in these plants OR becasue they have been isolated from those plants?
-line 1010 A.agricola NOT Agricola
line 143 "indole" not inodle
Author Response
Please, check the attached file.
In the attached files, authors have replied to the reviewer 1 one by one.
All changes and revisions and corrections have shown in YELLOW highlight. Thanks a lot for important, useful and wise corrections, revisions, and suggestions.
Regards

Reviewer 2 Report
Comments and Suggestions for Authors
The article “Study of Azospirillum spp. Important Source of Phytohormones for Increase Growth, Productivity, Yield and Building Stress Resilience ” is devoted to the important and acute theme of protection of plants against stressors and increasing of its productivity with beneficial microbes. Authors provide and analyze a lot of information on this theme, but the title, which focuses on phytohormones, does not reflect the content of the manuscript, since the text includes a little information on phytohormones exactly. I think careful editing should be made to resolve this problem.
I have some comments:
1)Title: I recommend to centre around the main topic (maybe “Strains of the genus Azospirillum Increase Productivity and Stress Resilience in plants”???)
2)Grammatics must be checked carefully
3)Lines 70-74 should be removed, authors shouldn’t insert methods of the data collection in review papers.
4)Authors must specify full systematic names of species then they are mentioned for the first time (for example A. brasilense Tarrand, Krieg & Döbereiner, 1978) in the main text and tables.
5)All strains and isolates definitions must be given (for example, Azospirillum brasilense Az39 or A. brasiliense sp. if it cannot be possible)
6)In all tables sentences on strains properties must be rewritten (for example “It could increase rosettes diameter and seed yield” should be replaced on “increase of rosettes diameter and seed yield”)
7)Table 1 and Table 3 should be combined.
8)Table 2 must be rebuilded, columns and parameters must be brought into line with tables 1, 3. Examples of the increase of stress tolerance under the influence of Azospirillum strains (references 180-183, 90, 208 etc.) must be transferred from tables 1, 3 to the table 2.
9)Data given in the tables must be discussed in the text and analysed in some extents (Sentences “The effects of inoculation with different Azospirillum strains on the yield of some of the most important agricultural and horticultural plants in various types of stress are presented in Table 1” and “The participation of Azospirillum in plants, defense against stress factors are shown in Table 2” are not enough).
Author Response
We are thanking you for important corrections, and revision. We have improved all parts, and we also revise the manuscript and add complementary information.
Please, check the attached file.

Round 2
Reviewer 1 Report
Comments and Suggestions for Authors
no more comments
Author Response
Dear Respected Editor
Greetings. We are thanking you for important comments and wise suggestions.
We also thanking you for accepting our article and also accept the revised version of the manuscript.
Thanks a lot again for your attentions and considerations.
Best Regards
Reviewer 2 Report
Comments and Suggestions for Authors
Unfortunately, a part of my comments were ignored by the authors of the manuscript in spite of their positive replies on my questions.
4)Authors must specify full systematic names of species then they are mentioned for the first time (for example A. brasilense Tarrand, Krieg & Döbereiner, 1978) in the main text and tables.
Reply: We are thanking you for the correction. We have corrected and written the full name and it has shown with Yellow highlight in the text.
Authors didn’t specify full systematic names of all species, mentioned in the manuscript, except A. brasilense Tarrand, Krieg & Döbereiner, 1978. And I insist that they must give it to all species of bacteria and plants.
5) All strains and isolates definitions must be given (for example, Azospirillum brasilense Az39 or A. brasiliense sp. if it cannot be possible)
Reply: Thanks a lot for your comment. We have improved all parts and all tables accordingly, however, in some articles, the specific names of Azospirillum was not available, and in some articles, just ABSTRACT of articles were available, so we had to just write the common name, however, we have done big improvement and changed in the manuscript.
Authors didn’t give strains and isolates names, a random check of its availability in literature sources showed that it is possible in most causes. Authors must improve it.
7)Table 1 and Table 3 should be combined.
Reply: We appreciate your comment and respect you. But, these two tables are completely different. For example, in Table 1, we have analyzed and studied the influence of inoculation with different strains of Azospirillum on the yield of some horticultural and agricultural plants with considering different stress types such as Drought, Cadmium, and some other minerals, Salinity, etc., while in Table 3, we have focused on the whole parameters and topics, with special focus on the effects of different species on yield and yield components.
I don’t see any reasons for differentiation of data in tables 1 and 3. Authors must combine these data in the one table, include plant species in all cases and describe the type of the stress in column Key points, for example “Increase shoot fresh weight under the cadmium pollution”
8)Table 2 must be rebuilded, columns and parameters must be brought into line with tables 1, 3. Examples of the increase of stress tolerance under the influence of Azospirillum strains (references 180-183, 90, 208 etc.) must be transferred from tables 1, 3 to table 2.
Authors didn’t rebuild Table 2. If it’s so difficult, they can include at least plant species or families on which the effects of bacterial species were demonstrated.
Author Response
Dear Respected Reviewer 2
We are thanking you for the comments. Please, check the attached files which is the reply to the Reviewer. We have tried and searched all the sources that you have mentioned, we sincerely think that the article can not be revised better on the basis of your useful and fruitful comments.
We are thanking you and your team and valuable suggestions once again.
All changes are shown in Yellow color in the manuscript.
Regards

Round 3
Reviewer 2 Report
Comments and Suggestions for Authors
Authors made great work on the manuscript refinement. I have received satisfactory answers on all my comments, but authors must prepare minor revisions of the manuscript.
1) Delete the word “strain” in all cases (for example, “A. argentinense strain Az19” should be replaced on “A. argentinense Az19” and all such examples in tables and the text)
2) Authors must specify full systematic names of plant species when they are mentioned for the first time too. In all cases in tables and the text.
Author Response
Dear Respected Reviewer 2
Greetings. We are thanking you again for the correction and revision. The correction has done accordingly. All changes in the manuscript are shown in Yellow highlight. We are thanking you again. Please, also check the attached file.
Best Regards
